# Asymptotic Properties for Bayesian Neural Network in Besov Space

**Kyeongwon Lee**
Department of Statistics
Seoul National University
Seoul, Republic of Korea 08826
lkw1718@snu.ac.kr

**Jaeyong Lee**
Department of Statistics
Seoul National University
Seoul, Republic of Korea 08826
leejyc@gmail.com

## Abstract

Neural networks have shown great predictive power when applied to unstructured data such as images and natural languages. The Bayesian neural network captures the uncertainty of prediction by computing the posterior distribution of the model parameters. In this paper, we show that the Bayesian neural network with spike-and-slab prior has posterior consistency with a near minimax optimal convergence rate when the true regression function belongs to the Besov space. The spike-and-slab prior is adaptive to the smoothness of the regression function and the posterior convergence rate does not change even when the smoothness of the regression function is unknown. We also consider the shrinkage prior, which is computationally more feasible than the spike-and-slab prior, and show that it has the same posterior convergence rate as the spike-and-slab prior.

## 1 Introduction

The neural network is a machine learning technique that can handle complex structures in data by combining linear and non-linear transformations [Goodfellow et al., 2016]. The neural network has shown great predictive power when applied to unstructured data ranging from image data to natural languages. The neural network automatically grasps the inherent structure of the data and makes end-to-end learning possible only with inputs and outputs without explicit intermediate steps, which is the main reason for the rapid development of numerous artificial intelligence applications.

The Bayesian neural network (BNN) provides the uncertainty of the prediction from the probability distribution of the model parameters. Before observing the data, a prior distribution that reflects prior beliefs about the model is put over the parameter space of the neural network model. As observations are added, a posterior distribution is computed by updating the prior distribution via Bayes' rule. Neal [1996], Williams [1996] showed that a shallow BNN which has only one infinite-width hidden layer converges to a Gaussian process regression model. This means that the shallow BNN can be regarded as a finite approximation of the nonparametric Bayesian regression model. Lee et al. [2017], Matthews et al. [2018] extended this result to the deep BNN model.

However, it is difficult to compute the exact posterior distribution of the BNN. Various methods have been proposed for BNN through approximate Bayesian computation algorithms. The methods based on the Markov chain Monte Carlo (MCMC) algorithm such as the Gibbs sampler [Geman and Geman, 1984, Casella and George, 1992] and Hamiltonian Monte Carlo (HMC) [Neal et al., 2011] are effective to approximate true posterior. By constructing a Markov chain that has the desired distribution as its stationary distribution, one can obtain a sample of the desired distribution. Recently, scalable MCMC algorithms such as stochastic gradient Langevin dynamics [Welling and Teh, 2011] and stochastic gradient HMC (SGHMC, Chen et al. [2014]) were proposed. The variational inference (VI) is an approximation method that finds the closest variational distribution from the posterior

36th Conference on Neural Information Processing Systems (NeurIPS 2022).

distribution. VI method changes the Bayesian inference problem to an optimization problem. Graves [2011] proposed to use of a Gaussian distribution as variational distribution for BNN and Kingma and Welling [2013] suggested the *reparameterization trick* for Gaussian variational distribution. Practically, VI methods for BNN are faster and more stable than MCMC methods. Blundell et al. [2015] suggested *Bayes by Backprop* algorithm for BNN via variational inference. There are also studies that approximate BNN with the dropout [Gal and Ghahramani, 2016] or the Gaussian process [Lee et al., 2017].

## 1.1 Related works

The performance of the neural network is justified by the *universal approximation* capability of the neural network. It is known that even a shallow neural network can approximate arbitrary continuous functions sufficiently precisely in $L^p$-sense [Cybenko, 1989, Hornik, 1991], and as the network deepens, the expressive power of the neural network gets stronger [Delalleau and Bengio, 2011, Bianchini and Scarselli, 2014, Telgarsky, 2015, 2016]. Lee III [1999] showed that the BNN has the universal approximation property.

As a measure of the expressive power of a model, consider the range of function spaces that a model can express. The choice of the activation function can affect the expressive power of a model, and in this paper, we consider the Rectified Linear Unit (ReLU) $ReLU(x) = \max\{0, x\}$ [Nair and Hinton, 2010], which is the most popular activation function in practice. Yarotsky [2017] evaluated the approximation error of the neural network using the ReLU activation function (ReLU network) for functions in the Hölder space. Schmidt-Hieber [2020] showed that the regularized least squared estimator performed by the ReLU network in a nonparametric regression problem converges to the true regression function with nearly minimax rate up to logarithmic factors on the Hölder space. Suzuki [2018] extended the results in Schmidt-Hieber [2020] from the Hölder space to the Besov space which contains the Hölder and Sobolev spaces. Polson and Ročková [2018] proved a Bayesian version of the results in Schmidt-Hieber [2020]. They proved the Bayesian ReLU network with spike-and-slab prior has posterior consistency, i.e., the posterior probability is concentrated on the true parameter value as the number of observations increases to infinity, when the priors on the parameters of the ReLU network are spike-and-slab priors. Chérief-Abdellatif [2020] showed that the results in Polson and Ročková [2018] are still valid when variational methods are applied.

All of the above results assume a sparse structure in the neural networks. Sparsity is often assumed in high-dimensional models for asymptotic results. Castillo et al. [2015] showed theoretical properties including posterior consistency and rates of consistency of the high-dimensional linear regression model under the spike-and-slab prior. Song and Liang [2017] extended the results in Castillo et al. [2015] to the shrinkage priors which are more feasible than the spike-and-slab prior.

## 1.2 Main contribution

We show that the posterior of the Bayesian ReLU network with a spike-and-slab prior with appropriate width, depth, and sparsity level is consistent with a near minimax rate. We extend the result of Suzuki [2018] to the Bayesian framework and the results of Polson and Ročková [2018] to the Besov space. As in Polson and Ročková [2018], we show that the same convergence rate is achieved even when the smoothness parameter of the true regression function is not known. That is, the ReLU network has rich expressive power, and this ability is not constrained even without the smoothness information of the true function. We get a similar result for the shrinkage prior.

We try to narrow the gap between the study of the theoretical properties and the practical application of the Bayesian ReLU networks. For instance, Gaussian distribution is frequently considered as a prior distribution in practical Bayesian ReLU networks. In this case, the posterior distribution may not be consistent since the posterior becomes heavy tail distribution over the parameter space [Vladimirova et al., 2019]. Approaches to interpreting the Bayesian neural networks as Gaussian Process [Lee et al., 2017] avoid this problem by setting the variance of the prior distribution to be proportional to the width of the network. We suggest similar sufficient conditions for the posterior consistency of the Bayesian ReLU networks. Compared to precedent studies, our setting is closer to practical use. The results in Suzuki [2018] and Polson and Ročková [2018] require to solve the optimization problem under the $l_0$ constraint or compute posterior from spike-and-slab prior respectively. We suggest a BNN model, which is easy to infer and consistent with the (nearly) optimal convergence rate.

The rest of the paper is organized as follows. We set up the problem and introduce the necessary concepts in Section 2. We present the obtained theoretical results in Section 3 and numerical examples in Section 4. A summary of the paper and a discussion are given in Section 5.

## 2 Preliminaries

### 2.1 Notation

The floor function $\lfloor x \rfloor$ denotes the largest integer less than or equal to $x \in \mathbb{R}$, the ceiling function $\lceil x \rceil$ denotes the smallest integer greater than or equal to $x \in \mathbb{R}$, and $\|a\|_p$ is an $l_p$-norm of a real vector $a$. The $u$-th derivative of a $d$-dimensional function $f$ is denoted by $D^u f$ with $u \in \{0, 1, \cdots\}^d$. The covering number $N(\epsilon, A, d)$ is the minimal number of $d$-balls of radius $\epsilon > 0$ necessary to cover a set $A$. We denote the dirac-delta function at $a \in \mathbb{R}$ by $\delta_a$. We use $A \lesssim B$ as shorthand for the inequality $A \leq CB$ for some constant $C > 0$ and $A \vee B$ as shorthand for the $\max\{A, B\}$.

### 2.2 Model

Suppose that $n$ input-output observations $\mathbb{D}_n = (X_i, y_i)_{i=1}^n \subset [0,1]^d \times \mathbb{R}$ are independent random sample from a regression model

$$y_i = f_0(X_i) + \xi_i \ (i = 1, 2, \cdots, n), \tag{1}$$

where $(\xi_i)_{i=1}^n$ is an i.i.d. sequence of Gaussian noises $\mathcal{N}(0, \sigma^2)$ with known variance $\sigma^2 > 0$ and $f_0$ is the true regression function belonging to the space $\mathscr{F}$.

In frequentist statistics, the maximum likelihood estimation is used to estimate the function $f_0$. From the machine learning viewpoint, it is equivalent to finding $\hat{f}$ that minimizes the square loss. Suzuki [2018] showed that $f_0$ can be estimated with a nearly optimal rate in this setting.

In practice, a sparse regularization term $r(f)$ such as dropout and $l_1$-regularizer is also considered together with a loss function. This is equivalent to estimating the MAP (maximum a posteriori) in the Bayesian modeling. The posterior distribution gives not only predictive values but also the uncertainty of the prediction. Thus, in the Bayesian statistics, it is important to find sufficient conditions for a prior distribution or a regularizer which guarantees the consistency or optimal convergence rate of the posterior.

### 2.3 Function spaces

In this section, we introduce function spaces that statisticians have studied for the *smoothness* of functions. Let $\Omega = [0,1]^d \subset \mathbb{R}^d$ be the domain of functions we consider.

The $L^p$-norm of a function is defined as follows:

$$\|f\|_{L^p} := \begin{cases} \left(\int_\Omega |f(x)|^p dx\right)^{1/p} & 0 < p < \infty, \\ \sup_{x \in \Omega} |f(x)| & p = \infty \end{cases}$$

where $f$ is a real-valued function defined on $\Omega$. The $L^p$ space is the function space consisting of functions with bounded $L^p$-norm and is denoted by $L^p(\Omega) = \{f : \|f\|_{L^p} < \infty\}$.

Let $s > 0$ be the smoothness parameter and $m = \lfloor s \rfloor$. The $s$-Hölder norm of function $f$ is defined by

$$\|f\|_{C^s} := \sup_{\|u\|_1 \leq m} \|D^u f\|_{L^\infty} + \sup_{\|u\|_1 = m} \sup_{x,y \in \Omega} \frac{|D^u f(x) - D^u f(y)|}{|x - y|^{s-m}}.$$

The $s$-Hölder space $C^s(\Omega)$ is defined as a set of functions with bounded $s$-Hölder norm. We call $s$ the smoothness parameter of the $s$-Hölder space. Let $k \in \mathbb{N}$ and $1 \leq p \leq \infty$. The Sobolev norm of a function $f$ is defined by

$$\|f\|_{W^{k,p}} := \begin{cases} \left(\sum_{\|\alpha\|_1 \leq k} \|D^\alpha f\|_{L^p}^p\right)^{1/p} & 1 \leq p < \infty \\ \sup_{\|\alpha\|_1 \leq k} \|D^\alpha f\|_{L^\infty} & p = \infty. \end{cases}$$

The Sobolev space $W^{k,p}(\Omega)$ is defined by the function space consisting of functions with bounded Sobolev norm.

Note the notion of the smoothness of a function is related to the differentiability of the function. The Besov space extends the concept of smoothness. Before defining the Besov space, define the $r$-th modulus of smoothness of the function $f$ as follows [Gine and Nickl, 2016]:

$$w_{r,p}(f,t) = \sup_{\|h\|_2 \leq t} \|\Delta_h^r(f)\|_p, \tag{2}$$

$$\Delta_h^r(f)(x) = \begin{cases} \sum_{j=0}^r \binom{r}{j}(-1)^{r-j}f(x+jh) & x \in \Omega, x+rh \in \Omega, \\ 0 & \text{o.w.} \end{cases} \tag{3}$$

For $0 < p, q \leq \infty$, $s > 0$, $r = \lfloor s \rfloor + 1$, define the Besov norm of a function $f$ by

$$\|f\|_{B_{p,q}^s} := \|f\|_p + \begin{cases} \left(\int_0^\infty (t^{-s}w_{r,p}(f,t))^q \frac{dt}{t}\right)^{1/q} & q < \infty, \\ \sup_{t>0}\{t^{-s}w_{r,p}(f,t)\} & q = \infty. \end{cases} \tag{4}$$

The Besov space $B_{p,q}^s(\Omega)$ is defined as the set of functions with finite Besov norms [Gine and Nickl, 2016], i.e.,

$$B_{p,q}^s(\Omega) = \left\{ f : \|f\|_{B_{p,q}^s} < \infty \right\}. \tag{5}$$

Note that the Besov spaces can be defined without the differentiability and continuity of functions, and are more general than the Hölder and Sobolev spaces, which are subspaces of the Besov spaces. The Besov space may contain complicated functions including discontinuous function if $d/p \geq s$. For instance, the Cantor function $f_1$ does not belong to any Sobolev space since it does not have a weak derivative but belongs to a Besov space [Sawano, 2018]. In fact, $f_1 \in C^{\log 2/\log 3}([0,1]) \subset B_{\infty,\infty}^{\log 2/\log 3}([0,1])$. Also,

$$f_2(x) = \begin{cases} 1/\log(x/2), & 0 < x \leq 1 \\ 0, & x = 0 \end{cases} \tag{6}$$

is not in any Hölder space. Note when $p < 2$, the smoothness of the Besov space is inhomogeneous [Suzuki, 2018]. Donoho and Johnstone [1994] suggested spatially variable functions including Blocks

$$f_3(x) = \sum_j h_j K(x - x_j), \; K(x) = (1 + \text{sgn}(x))/2,$$
$$(x_j) = (0.1, 0.13, 0.15, 0.23, 0.25, 0.40, 0.44, 0.65, 0.76, 0.78, 0.81), \tag{7}$$
$$(h_j) = (4, -5, 3, -4, 5, -4.2, 2.1, 4.3, -3.1, 2.1, -4.2),$$

where $\text{sgn}(x)$ is a function that extracts the sign of a real number $x$, and HeaviSine

$$f_4(x) = 4\sin(4\pi x) - \text{sgn}(x - 0.3) - \text{sgn}(0.72 - x). \tag{8}$$

Since $f_2$, $f_3$ and $f_4$ are functions of bounded variation, in $B_{1,\infty}^1([0,1])$ [Peetre, 1976, Suzuki, 2018]. We plot the functions $f_1$, $f_2$, $f_3$ and $f_4$ in Figure 1.

## 2.4 Sparse neural network

Define (deep) neural network space generated by a parameter space $\Theta$ as follows:

$$\Phi(\Theta) = \left\{ f_\theta(x) = (W^{(L+1)}(\cdot) + b^{(L+1)}) \circ \zeta \circ \cdots \circ \zeta \circ (W^{(1)}x + b^{(1)}) : \right.$$
$$\left. \theta = (W^{(1)}, b^{(1)}, \cdots, W^{(L+1)}, b^{(L+1)}) \in \Theta \right\}. \tag{9}$$

Figure 1: Example functions in the Besov spaces

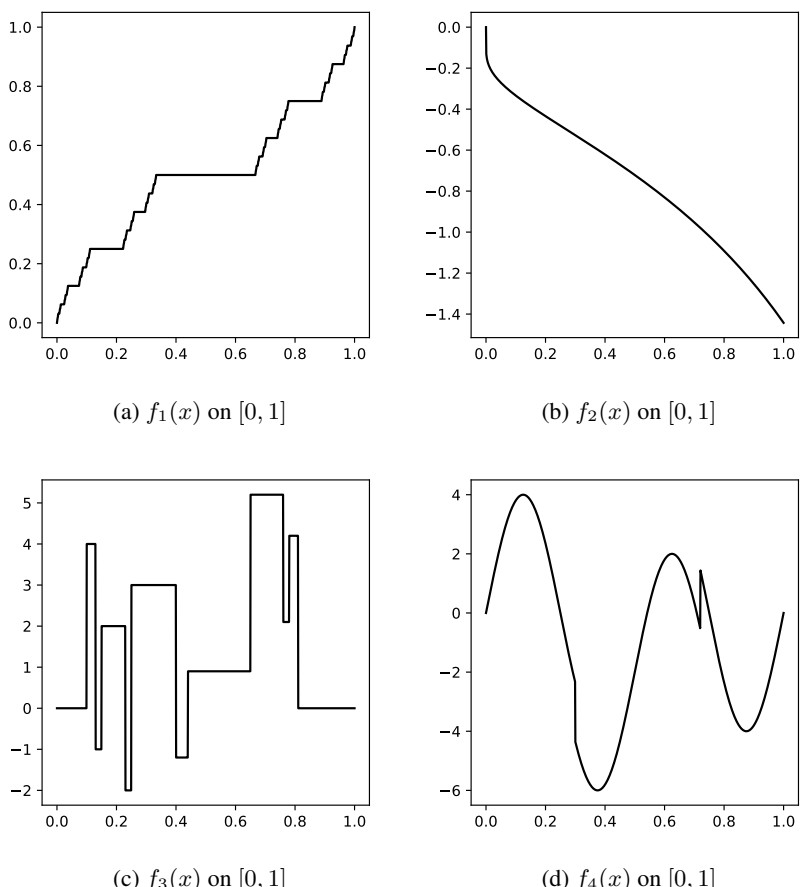

(a) $f_1(x)$ on $[0,1]$

(b) $f_2(x)$ on $[0,1]$

(c) $f_3(x)$ on $[0,1]$

(d) $f_4(x)$ on $[0,1]$

where $\zeta$ is an activation function. In this paper, we consider $\zeta(x) = ReLU(x)$ and the following $B$-bounded and $S$-sparse parameter space [Suzuki, 2018]

$$
\begin{aligned}
\Theta(L, W, S, B) := \Big\{ & W^{(l)} \in \mathbb{R}^{p_{l-1} \times p_l}, b^{(l)} \in \mathbb{R}^{p_l}, \\
& p_l = W, l = 1, 2 \cdots, L, \ p_0 = d, p_{L+1} = 1, \\
& \|\theta\|_0 = \sum_{l=1}^{L+1} \left( \left\|W^{(l)}\right\|_0 + \left\|b^{(l)}\right\|_0 \right) \le S, \\
& \|\theta\|_\infty = \max_l \left\{ \left\|W^{(l)}\right\|_\infty \vee \left\|b^{(l)}\right\|_\infty \right\} \le B \Big\}
\end{aligned}
\tag{10}
$$

and denote $S$-sparse neural network space with $B$-bounded parameters $\Phi(\Theta(L, W, S, B))$ as $\Phi(L, W, S, B)$. We can define the entire neural network model space as follows

$$
\Phi = \bigcup_{L=1}^{\infty} \bigcup_{W=1}^{\infty} \bigcup_{S=0}^{\infty} \bigcup_{B=0}^{\infty} \Phi(L, W, S, B).
\tag{11}
$$

## 3 Main results

Let $\mathcal{UB}(F) = \left\{ f \in L^\infty([0,1]^d) : |f(x)| \le F, \ \forall x \in [0,1]^d \right\}$ be an uniformly bounded family of functions with bound $F > 0$. Consider $\mathscr{F} = B_{p,q}^s([0,1]^d) \cap \mathcal{UB}(F)$. For each $n \in \mathbb{N}$, and $f \in \mathcal{F}$,

let $P_f^{(n)}$ be the probability measure of the data $\mathbb{D}_n = (X_i, y_i)_{i=1}^n$ with a density function $p_f^{(n)}$ with respect to the measure $\mu^{(n)}$ and $\Pi$ be a prior distribution of $f$. The posterior distribution is given by

$$\Pi(A|\mathbb{D}_n) = \frac{\int_A p_f^{(n)}(\mathbb{D}_n) d\Pi(f)}{\int_{\mathcal{F}} p_f^{(n)}(\mathbb{D}_n) d\Pi(f)}, \tag{12}$$

for $A \subset \Theta$. Let

$$\|f\|_n = \left( \frac{1}{n} \sum_{i=1}^n (f(X_i))^2 \right)^{1/2}$$

be the norm on empirical measure $\mathbb{P}_n^X = \frac{1}{n} \sum_{i=1}^n \delta_{X_i}$ and

$$\|f\|_{L^2(P_X)} = \left( \int (f(X))^2 dP_X \right)^{1/2},$$

where $P_X$ is a marginal distribution of $X$. Suppose that $P_X$ has a bounded density function $p_X(x)$ on $[0, 1]^d$. Without loss of generality, we assume that $0 \le p_X(x) \le R \le 2$.

Suppose that $0 < F < \infty$, $0 < p, q \le \infty$, $\omega := d(1/p - 1/2)_+ < s < \infty$ and set $\nu = (s - \omega)/(2\omega)$. Assume that $m \in \mathbb{N}$ satisfies $0 < s < \min\{m, m - 1 + 1/p\}$. Let $N_n = \lceil n^{d/(2s+d)} \rceil$, $W_0 = 6dm(m + 2) + 2d$ and

$$L_n = L(N_n), \quad W_n = N_n W_0,$$
$$S_n = (L_n - 1) W_0^2 N_n + N_n, \quad B_n = O(N_n^\Xi), \tag{13}$$

where $c_{(d,m)} = \left( 1 + 2de \frac{(2e)^m}{\sqrt{m}} \right)^{-1}$, $L(N_n) = 3 + 2\lceil \log_2 \left( \frac{3^{d \vee m}}{\tau_n c_{(d,m)}} \right) + 5 \rceil \lceil \log_2(d \vee m) \rceil$, $\tau_n = N_n^{-s/d - (\nu^{-1} + d^{-1})(d/p - s)_+} (\log N_n)^{-1}$ and $\Xi = (\nu^{-1} + d^{-1})(1 \vee (d/p - s)_+)$.

Note the theorems in this section are similar to the results in Suzuki [2018], which studied the asymptotic properties of the least squared estimator. They showed that for the least squared estimator

$$\hat{f} = \underset{\bar{f}: f \in \Phi(L_n, W_n, S_n, B_n)}{\arg\min} \sum_{i=1}^n (y_i - \bar{f}(x_i))^2,$$
$$\mathbb{E}_{\mathbb{D}_n} \left[ \left\| f_0 - \hat{f} \right\|_{L^2(P_X)} \right] \lesssim n^{-\frac{2s}{2s+d}} (\log n)^3, \tag{14}$$

where $\bar{f}$ is the clipping of $f$ defined by $\bar{f} = \min\{\max\{f, -F\}, F\}$ for $f_0 \in \mathcal{UB}(F)$ and $\mathbb{E}_{\mathbb{D}_n}[A]$ is an expectation of $A$ with respect to the training data $\mathbb{D}_n$. Note the minimax convergence rate of a Besov function $f_0$ in (1) has a lower bound $n^{-\frac{s}{2s+d}}$ [Gine and Nickl, 2016, Donoho and Johnstone, 1998]. Donoho and Johnstone [1998] also showed that the minimax rate of a linear estimator has a lower bound, $n^{-\frac{s-(1/p-1/2)_+}{2s+1-2(1/p-1/2)_+}}$ for $d = 1$, $s > 1/p$ and $1 \le p, q \le \infty$ or $s = p = q = 1$. Thus, any linear estimator cannot achieve the minimax optimal rate on the Besov space for $p < 2$ and is dominated by the neural network model.

**Spike-and-slab prior**

In this subsection, we extend Theorem 6.1 in Polson and Ročková [2018]. They showed that the Bayesian ReLU networks are consistent with near minimax rates in the Hölder space. We extend the function space from the Hölder space to the Besov space which contains both Hölder and Sobolev spaces. Assume the following spike-and-slab prior $\pi(\theta)$

$$\pi(\theta_j | \gamma_j, L, W, S, B) = \gamma_j \tilde{\pi}(\theta_j | L, W, S, B) + (1 - \gamma_j) \delta_0(\theta_j),$$
$$\pi(\boldsymbol{\gamma} | L, W, S, B) = \frac{1}{\binom{T}{S}}, \tag{15}$$

$$\pi(L = L_n) = \pi(W = W_n) = \pi(S = S_n) = \pi(B = B_n) = 1, \tag{16}$$

where $\tilde{\pi}(\theta_j | L, W, S, B) = U(\theta_j; [-B, B])$ and $T = |\Theta(L, W, S, B)|$.

**Theorem 1.** *Assume model (1) and prior distribution (15) and (16). Suppose that $0 < F < \infty$, $0 < p, q \leq \infty$ and $d(1/p - 1/2)_+ < s$. Then the posterior distribution concentrates at the true function with a rate $\epsilon_n = n^{-s/(2s+d)}(\log n)^{3/2}$. That is,*

$$\Pi(f_\theta \in \Phi \cap \mathcal{UB}(F) : \|f_\theta - f_0\|_n > M_n \epsilon_n | \mathbb{D}_n) \to 0$$

*in $P_{f_0}^{(n)}$-probability as $n \to \infty$ for any $M_n \to \infty$.*

*Proof.* See appendix C.4. □

As mentioned in Suzuki [2018], the condition $d(1/p - 1/2)_+ < s$ indicates that $f_0 \in B_{p,q}^s([0,1]^d)$ is in $L^2([0,1]^d)$ and can be discontinuous. When $p = q = \infty$, $B_{p,q}^s([0,1]^d) = C^s([0,1]^d)$ and the result by Polson and Ročková [2018] is a special case of Theorem 1.

### Adaptive estimation

Theorem 2 below is an extension of Theorem 6.2 in Polson and Ročková [2018] to the Besov space. They showed that the Bayesian ReLU networks adapt to smoothness on the Hölder space under the spike-and-slab prior.

**Theorem 2.** *Assume model (1). Let*

$$\tilde{L}_n(H) = \lceil H(\log n) \rceil \vee 1, \ \tilde{W}_n(H, N) = HN, \ \tilde{S}_n(H, N) = HN\tilde{L}_n(H), \ \tilde{B}_n(H, N) = N^H. \tag{17}$$

*Consider the following prior*

$$N \stackrel{d}{=} 1 \vee \lceil Z/(\log n)^2 \rceil, \quad \pi_Z(Z) = \frac{\lambda_N^Z}{Z!(e^{\lambda_N} - 1)} \quad for \ Z = 1, 2, \cdots, \tag{18}$$

*and prior distribution (15) of $\theta$ given $\tilde{L}_n(H_n)$, $\tilde{W}_n(H_n, N)$, $\tilde{S}_n(H_n, N)$, $\tilde{B}_n(H_n, N)$ on the function space*

$$\Phi = \bigcup_{n=1}^{\infty} \bigcup_{N=1}^{\infty} \Phi(\tilde{L}_n(H_n), \tilde{W}_n(H_n, N), \tilde{S}_n(H_n, N), \tilde{B}_n(H_n, N)) \tag{19}$$

*for any $H_n \to \infty$ and $\lambda_N > 0$. Suppose that $0 < F < \infty$, $0 < p, q \leq \infty$ and $d(1/p - 1/2)_+ < s < \min\{m, m - 1 + 1/p\}$. The posterior distribution concentrates at the true function with a rate $\epsilon_n = n^{-s/(2s+d)}(\log n)^{3/2}$. That is,*

$$\Pi(f_\theta \in \Phi \cap \mathcal{UB}(F) : \|f_\theta - f_0\|_n > M_n \epsilon_n | \mathbb{D}_n) \to 0$$

*in $P_{f_0}^{(n)}$-probability as $n \to \infty$ for any $M_n \to \infty$.*

*Proof.* See appendix C.5. □

In Theorem 1, the prior distribution depends on the smoothness parameters $p$, $q$, and $s$. Theorem 2 shows that, as in Polson and Ročková [2018], even if the values of these parameters are unknown, the same posterior convergence rates can be obtained by considering the appropriate prior distribution.

### Shrinkage prior

Asymptotic properties of the spike-and-slab prior were obtained in Theorem 1, but its practical use for large neural network models is unsuitable due to the computational cost. The following prior distributions are computationally more feasible than the spike-and-slab prior.

The result to be introduced in this section shows that all the aforementioned results still hold even under the shrinkage prior. Assume the following prior

$$\pi(\theta|L, W, S, B) = \prod_{j=1}^{T} g(\theta_j|L, W, S, B) \tag{20}$$

where $T = |\Theta(L, W, S, B)|$, $g(t) := g(t|L, W, S, B)$ is a symmetric density function and decreasing on $t > 0$.

**Theorem 3.** *Assume model (1), prior distribution (16) and (20). Suppose that $0 < F < \infty$, $0 < p, q \leq \infty$ and $d(1/p - 1/2)_+ < s < \min\{m, m - 1 + 1/p\}$. Let $\epsilon_n = n^{-s/(2s+d)}(\log n)^{3/2}$ and $g(t)$ be a function such that*

$$a_n \leq \frac{\epsilon_n}{72 L_n (B_n \vee 1)^{L_n - 1}(W_n + 1)^{L_n}}$$

$$u_n = \int_{[-a_n, a_n]} g(t | L_n, W_n, S_n, B_n) dt \tag{21}$$

*satisfies*

$$\frac{S_n}{T_n} > 1 - u_n \geq \frac{S_n}{T_n} \eta_n, \tag{22}$$

$$-\log g(B_n | L_n, W_n, S_n, B_n) \lesssim (\log n)^2, \tag{23}$$

*continuous on $[-B_n, B_n]$ and*

$$v_n = \int_{[-B_n, B_n]^c} g(t | L_n, W_n, S_n, B_n) dt = o\left(e^{-K_0 n \epsilon_n^2}\right), \tag{24}$$

*where $\eta_n = \exp(-K n \epsilon_n^2 / S_n)$ for some $K$, $K_0 > 4$. The posterior distribution concentrates at the true function with a rate $\epsilon_n$. That is,*

$$\Pi(f_\theta \in \Phi \cap \mathcal{UB}(F) : \|f_\theta - f_0\|_n > M_n \epsilon_n | \mathbb{D}_n) \to 0$$

*in $P_{f_0}^{(n)}$-probability as $n \to \infty$ for any $M_n \to \infty$.*

*Proof.* See appendix C.6. □

Note that condition (22) is a continuous approximation for the spike part of (15). The next condition (23) means that the prior should have enough thick tail to sample true function. The last condition (24) restricts the thickness of the tail to prevent a function from being divergent.

Unlike the other priors, the shrinkage prior is relatively straightforward to implement. For example, to implement the model in Theorem 1, one can consider the MCMC algorithm [Sun et al., 2022] which combines Gibbs sampler, the Metropolis-Hastings algorithm, and the SGHMC algorithm. The shrinkage prior avoids the posterior computation with varying dimensions, and enables feasible computation.

Note that the conditions are sufficient conditions for the neural network model necessary to estimate the function in the worst case. In other words, we may infer true function $f_0$ sufficiently even for a prior distribution that satisfies only some conditions.

**Example 1** (Gaussian prior)**.** *Let $\epsilon_n = n^{-s/(2s+d)}(\log n)^{3/2}$ and $\psi$ be a density functions of the standard Gaussian distribution $\mathcal{N}(0, 1)$.*

$$g(t) = \frac{1}{\sigma_n} \psi\left(\frac{t}{\sigma_n}\right), \quad \sigma_n^2 = \frac{B_n^2}{2 K_0 n \epsilon_n^2}$$

*satisfies (24).*

**Example 2** (Gaussian mixture prior)**.** *Let $\epsilon_n = n^{-s/(2s+d)}(\log n)^{3/2}$, $\psi$ and $\Psi$ be a density and inverse survival functions of the standard Gaussian distribution $\mathcal{N}(0, 1)$ respectively.*

$$g(t) = \pi_{1n} \frac{1}{\sigma_{1n}} \psi\left(\frac{t}{\sigma_{1n}}\right) + \pi_{2n} \frac{1}{\sigma_{2n}} \psi\left(\frac{t}{\sigma_{2n}}\right)$$

*with*

$$\pi_{2n} = \frac{S_n}{T_n}, \; \pi_{1n} = 1 - \pi_{2n},$$

$$a_n \leq \frac{\epsilon_n}{72 L_n (B_n \vee 1)^{L_n - 1}(W_n + 1)^{L_n}}, \; \sigma_{2n}^2 < \frac{B_n^2}{2 K_0 n \epsilon_n^2}$$

*and*

$$a_n / \Psi\left(\frac{\pi_{2n}}{\pi_{1n}}\left[\frac{1}{2}\eta_n - \Psi^{-1}\left(\frac{a_n}{\sigma_{2n}}\right)\right]\right) \leq \sigma_{1n} < a_n / \Psi\left(\frac{\pi_{2n}}{\pi_{1n}}\left[\frac{1}{2} - \Psi^{-1}\left(\frac{a_n}{\sigma_{2n}}\right)\right]\right)$$

*satisfies (22) and (24).*

**Example 3** (Relaxed spike-and-slab prior). *Let $\epsilon_n = n^{-s/(2s+d)}(\log n)^{3/2}$, $\psi$ and $\Psi$ be a density and inverse survival functions of the standard Gaussian distribution $\mathcal{N}(0,\ 1)$ respectively.*

$$g(t) = \pi_{1n}\frac{1}{\sigma_{1n}}\psi\left(\frac{t}{\sigma_{1n}}\right) + \pi_{2n}U(t; -B_n,\ B_n)$$

*with*

$$\pi_{2n} = \frac{S_n}{T_n},\ \pi_{1n} = 1 - \pi_{2n},$$

$$a_n \le \frac{\epsilon_n}{72L_n(B_n \vee 1)^{L_n-1}(W_n+1)^{L_n}},$$

*and*

$$a_n/\Psi\left(\frac{\pi_{2n}}{2\pi_{1n}}\left[\frac{a_n}{B_n} - (1-\eta_n)\right]\right) \le \sigma_{1n} < a_n/\Psi\left(\frac{\pi_{2n}}{2\pi_{1n}}\frac{a_n}{B_n}\right)$$

*satisfies (22), (23) and (24).*

## 4 Numerical examples

As the sample size $n$ increases, the complexity of the neural network to obtain the theoretical optimality increases rapidly, which makes the results in the previous section limited for the practical application with large sample size. For instance, consider the estimation problems of Besov functions $f_1$, $f_2$, $f_3$ and $f_4$ with the Gaussian mixture prior with parameters as in Example 2:

$$B_n = 10N_n^{\min\{\Xi,1\}},\ K_0 = 5,\ \eta_n = \exp(-K_0 n\epsilon_n^2/S_n)$$

$$a_n = \frac{\epsilon_n}{72L_n(B_n \vee 1)^{L_n-1}(W_n+1)^{L_n}}$$

$$\sigma_{2n}^2 = \frac{B_n^2}{2(K_0+1)n\epsilon_n^2},\ \sigma_{1n} = a_n/\Psi\left(\frac{\pi_{2n}}{\pi_{1n}}\left[\frac{1}{2}\eta_n - \Psi^{-1}\left(\frac{a_n}{\sigma_{2n}}\right)\right]\right).$$

Tables 1 and 2 present the hyperparameters (model parameters) to estimate functions $f_i$, $i = 1,2,3,4$ with theoretical optimality. These values are affected by the complexity of the true regression function.

Table 1: Hyperparameters of BNN to estimate $f_1 \in B_{\infty,\infty}^{\log 2/\log 3}([0,1])$.

| $n$ | $L$ | $W$ | $\sigma_{1n}$ | $\sigma_{2n}$ | $\pi_{1n}$ | $\pi_{2n}$ |
|---|---|---|---|---|---|---|
| 100 | 39 | 400 | $3.747 \times 10^{-179}$ | $8.443 \times 10^{-1}$ | $8.719 \times 10^{-1}$ | $1.280 \times 10^{-1}$ |
| 1,000 | 43 | 1,100 | $1.505 \times 10^{-234}$ | $7.597 \times 10^{-1}$ | $9.534 \times 10^{-1}$ | $4.651 \times 10^{-2}$ |
| 10,000 | 45 | 2,950 | $1.508 \times 10^{-283}$ | $7.954 \times 10^{-1}$ | $9.826 \times 10^{-1}$ | $1.733 \times 10^{-2}$ |

Table 2: Hyperparameters of BNN to estimate $f_2$, $f_3$ and $f_4 \in B_{1,1}^1([0,1])$.

| $n$ | $L$ | $W$ | $\sigma_{1n}$ | $\sigma_{2n}$ | $\pi_{1n}$ | $\pi_{2n}$ |
|---|---|---|---|---|---|---|
| 100 | 41 | 250 | $8.669 \times 10^{-172}$ | $6.779 \times 10^{-1}$ | $7.957 \times 10^{-1}$ | $2.042 \times 10^{-1}$ |
| 1,000 | 43 | 500 | $1.259 \times 10^{-205}$ | $5.028 \times 10^{-1}$ | $8.977 \times 10^{-1}$ | $1.022 \times 10^{-1}$ |
| 10,000 | 47 | 1,100 | $1.963 \times 10^{-256}$ | $4.894 \times 10^{-1}$ | $9.535 \times 10^{-1}$ | $4.642 \times 10^{-2}$ |

In practice, models with smaller values of $L$ and $W$ can estimate the true function with a sufficient precision. We present the numerical experiments with the four functions in Appendix D.

## 5 Conclusions

In this paper, we justify the theoretical properties of neural networks that are currently widely used. Specifically, we focused on the asymptotic properties of the Bayesian neural networks and extend the Hölder space results in Polson and Ročková [2018] to the Besov space. Theorem 1 shows that the

Bayesian neural network with a spike-and-slab prior has consistent posterior with a near minimax rate at the true Besov functions. Theorem 3 extends the result to the shrinkage prior. According to these theorems, the Bayesian neural networks have posterior consistency and (nearly) optimal rates by assuming proper prior. Theorem 2 shows that even if the smoothness of the true function is not known, the same optimality can be obtained by considering a prior on the architecture of the neural networks.

The results in this paper are extensions of those in Suzuki [2018] to the Bayesian neural networks. We suggested a shrinkage prior for the Bayesian neural network as a practical prior that can achieve the optimal rate while providing the uncertainty of the prediction. Suzuki [2018] showed the theoretical properties of the neural networks in the Besov space via nonparametric regression based on B-spline. We believe similar results can be obtained using the Bayesian LARK B-spline model [Park et al., 2021].

We obtained the theoretical results under the specific setting, fully connected networks with the ReLU activation function. As mentioned in Section 4, the conditions to obtain the theoretical optimality is stringent for practical applications with large sample sizes. In Theorem 3, we mentioned the conditions that the shrinkage prior must satisfy and depend on the model parameters $L_n$, $W_n$, and $B_n$. For adaptive estimation like Theorem 2, it is a necessary to propose a general shrinkage prior distribution that satisfies all conditions even when the parameters of the network are varying. The results of this study are just beginning and more theoretical research is needed for the Bayesian neural networks with industrial use, such as visual data analysis and natural language processing.

## Acknowledgments

Kyeongwon Lee and Jaeyong Lee were supported by the National Research Foundation of Korea (NRF) grant funded by the Korea government(MSIT) (No. 2018R1A2A3074973 and 2020R1A4A1018207).

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
