# OpenReview forum: "Asymptotic Properties for Bayesian Neural Network in Besov Space"
_NeurIPS.cc/2022/Conference — NeurIPS 2022 Accept_

### Official Review · Reviewer_x58D · 2022-07-03

**Rating:** 5
**Confidence:** 2
**Soundness:** 3 good
**Presentation:** 2 fair
**Contribution:** 3 good

**Summary:**

The paper proposes the theoretical results for Bayesian neural networks in the Besov space. It extends the results from the previous work in Holder space into the Besov space. It shows that Bayesian neural networks with the spike and slab priors and further relaxes it to be just shrinkage priors converge to the true regression function in the Besov space.

**Questions:**

My main concern with the paper is its clarity and I don’t have questions on that.

The other smaller concern is regarding the numerical example, so maybe the authors can answer questions about it, however, as this is not the major concern, the authors should be free to give priority to questions to other reviewers in their rebuttal if they would help to resolve some major concerns from the other reviewers.

1. Wouldn’t MCMC methods with true posterior be more appropriate for the purpose of the numerical example?
2. What can the authors say about the claims about practicality of the method?


**Limitations:**

There is no discussion on this, but since the work is theoretical it is not the major issue

**Strengths And Weaknesses:**

**Update after rebuttal:** I would like to thank the reviewers for their incredible job during the rebuttal and by improving the submission very much. I believe that the clarity of the paper has drastically improved and therefore I am raising my score. Even though after the correction of the theory the numerical example does not satisfy the theory now I believe the submission provides an interesting step towards better theoretical guarantees of BNNs.

=================================

**Originality:**
*Strengths:* The main results appear to be novel and an interesting extension of the previous results in Holder space into the Besov space.

**Quality:**
*Strengths:* I am sorry but I have to admit that I cannot assess the actual theoretical contribution of the paper as it is beyond my comfort level zone, but also because in contrast to some people who prefer formulae over word description I would prefer the latter to better understand the formulae. Please see comments on this below in Clarity section.

*Weaknesses:* Though I appreciate that the main contribution of the paper is theoretical, the numerical example provided is not too satisfying.

* The method is only used on 2 synthetic examples with analytical functions
* The main theoretical results are about the posterior, but the numerical experiment uses the Bayes by Backdrop for inference, which is implementation of the variational inference, i.e. approximation of the true posterior. Wouldn’t MCMC methods with true posterior be more appropriate for the purpose of the numerical example?
* There are several claims that the paper proposes a “practical” model, however, there is no empirical evidence on that. Some discussion on actual computational time used and how feasible the inference with the considered priors would be appreciated


**Clarity:**
*Strengths:* The text where available is mostly well written and easy to follow. The paper tries to tell the full story with a proper introduction rather than jumping straight into it as if a reader has just finished reading the previous work and doesn’t require any introduction - the approach used in many recent papers.

*Weaknesses:* There is not so much text in the paper but a lot of formulae. I appreciate there are people who prefer formulae over words, but there are others who much better read words over formulae. Word description of formulae would be much appreciated. Also, the notation is not very careful which does not help further with readability of this lot of maths.
If the space is an issue, some general paragraphs from the introduction can easily be reduced or removed to save some space. Figures 2 and 3 can also lose the histogram part to save some space. (The histogram part can go into the appendix also)

**Significance:**
*Strengths:* The paper addresses a very important problem of finding theoretical guarantees for deep learning, Bayesian deep learning in particular in this case. It does seem to make a step further in extending those guarantees (by providing them in a bigger Besov space).

**Summary:** Readability of the paper, namely the lack of words over maths, is the main issue with this submission in my opinion. So much so that it is difficult to assess the quality of the paper. It seems the paper has significant and original results but it should also be easier to understand these results while reading the paper. Moreover, the numerical example can be largely improved.

Specific comments/suggestions:
1. Line 10. “In other words” - a bit inappropriate use here as it hasn’t been said before anything about practicality, only about asymptotic properties
2. “Posterior consistency” is never properly defined in the paper
3. Line 96, “u \in {0, 1, …, d}”?
4. D^u is used for u-th derivative, and then D_n is used for a dataset. It is a bit confusing though technically it is not overlapping notation. It is probably better to use another letter for a dataset
5. Equation between lines 120 and 121. Both D^u and d^u is used. What is the difference?
6. Eq. (7) - \eta is not defined
7. Line 152, \delta is already used for the Dirac function
8. Theorem 1, M_n is not defined, how does it depends on n?
9. Line 204, N is already used, probably it is better to used \mathcal{N} for the normal distribution
10. Figures 2-3. Are dots training data points? Left column plots can used “x” and “y” labels for axis, also subplots without labels look too strange for me. It seems they may have “(a)”, “(b)”, … labels
11. Line 218. “shrinkage prior” - “spike-and-slab prior” was meant?
12. Lines 229-230, “Experiments … will also be of great help…” - not clear, is it suggested as future work? It would indeed be nice to have some comparison in the current submission.


Minor:
1. Line 21, “Before observing the data, A” - lower case for “A”
2. Line 36, “are” -> “were”
3. Line 69, “than THE spike-and-slab prior”
4. Line 82. “GP” - acronym is not introduced
5. Line 86, “to solve THE optimization problem”
6. Line 88, “with THE (nearly) optimal convergence rate”
7. Line 148, “Let” - lower case for “L”
8. Line 211, “were” -> “are”

---

> ### Author Response · Authors · 2022-08-02
> **Reply to reviewer x58D**
>
>
> Thanks for your comments and suggestions. First of all, we are sorry that our expression is unfriendly.
>
> ## Q.
>
> > Wouldn’t MCMC methods with true posterior be more appropriate for the purpose of the numerical example?
>
> > There are several claims that the paper proposes a “practical” model, however, there is no empirical evidence on that. Some discussion on actual computational time used and how feasible the inference with the considered priors would be appreciated
>
> Thank you for your suggestion. We changed the algorithm in numerical examples to NUTS algorithm (one of the Hamiltonian Monte Carlo algorithm) from the variational approximation.
>
> In addition, as mentioned in revised submission, we confirmed that some calculations of the model parameter proposed in the submitted paper were wrong and corrected them. The complexity of the model required for the experiments has grown, and it is difficult to implement it numerically. Thus, we replaced the numerical experiments to small model which shows interesting results though **it does not satisfy the conditions of the paper**.
>
> We plan to use SGHMC (Stocahstic gradient Hamiltonian Monte Carlo) as the algorithm for the real data examples. In fact, we already checked that the following papers suggest methods for extracting MCMC samples from the spike-and-slab prior distribution through the SGHMC algorithm.
>
> * Sun, Y., Song, Q., & Liang, F. (2022). Learning sparse deep neural networks with a spike-and-slab prior. Statistics & Probability Letters, 180, 109246.
> * Song, Q., Sun, Y., Ye, M., & Liang, F. (2020). Extended stochastic gradient Markov chain Monte Carlo for large-scale Bayesian variable selection. Biometrika, 107(4), 997-1004.
>
> We used the weight and bias parameters of the network model estimated by the frequentist method as the initial value of the NUTS algorithm, it will be possible to obtain a more efficient (nice and fast) MCMC sample than the above papers. This method would be applicable to large models such as ResNet. Unfortunately, due to lack of time, the experimental results were not obtained this submission. We would like to propose inferential algorithms together through follow-up studies.
>
> ## Q.
>
> > Line 10, “Posterior consistency”, and more:
>
> Thanks for your detailed comments. In the revised submission, we have corrected the ambiguous or insufficient description.
>
> ## Q.
>
> > Line 96, “u \in {0, 1, …, d}”?
>
> The part you pointed out is about 'with what component to differentiate', and the existing notation is correct. For instance, $D^u f(x)$ with $u=(1, 1, 1, ..., 1)$ means
> $$ \frac{\partial^d f(x)}{\partial x_1 \partial x_2 \cdots \partial x_d}.$$
>
> ## Q.
>
> > $D^u,~\delta$, $N$ for the normal distribution, and more:
>
> Thanks for your detailed comments. In the revised submission, we have corrected notations that may cause confusion.
>
> ## Q.
>
> > Posterior consistency, $\eta$ in Eq. (7), and more:
>
> Thanks for your detailed comments. In the revised submission, we have added explanations for undefined terms and symbols you mentioned.
>
> ## Q.
>
> > Theorem 1, M_n is not defined, how does it depends on n?
>
> $M_n$ means any sequence that is sufficiently large and increasing indefinitely as $n$ grows. Anything is fine as long as it goes infinity.
>
> ## Q.
>
> > Figures 2 and 3 can also lose the histogram part to save some space. (The histogram part can go into the appendix also)
> > Figures 2-3. Are dots training data points? Left column plots can used “x” and “y” labels for axis, also subplots without labels look too strange for me. It seems they may have “(a)”, “(b)”, … labels
>
> As you advised, we move the figures to the appendix and write description more detail than before. Thank you for your helpful advice.
>
> ## Q.
>
> > Line 218. “shrinkage prior” - “spike-and-slab prior” was meant?
>
> As you pointed out, that was a our mistake. Thanks for the detailed comments. We have corrected the text you mentioned in the revised submission.
>
> ## Q.
>
> > Lines 229-230, “Experiments … will also be of great help…” - not clear, is it suggested as future work? It would indeed be nice to have some comparison in the current submission.
>
> What you pointed out is correct. We are planning to compare with statistical models (in a narrow sense) such as Bayesian LARK B-spline model and Bayesian neural network model as a follow-up study. From a theoretical point of view, it has been shown that the models to be compared are (nearly) optimal, but more research is needed in a implementation to make comparisons with real data.
>
> ## Q.
>
> > Minor:
>
> Thanks for your detailed comments. We have corrected everything you mentioned in the revised submission.

---

> > ### Comment · Reviewer_x58D · 2022-08-03
> > **Thank you for your responses**
> >
> > Thank you very much for your detailed answer. You didn't have to go through answering my specific comments, but thank you that you did.
> >
> > Could you please elaborate a bit on "it does not satisfy the conditions of the paper"? Does this mean that the numerical examples are just a proxy now and there is no numerical evidence that supports your theory that satisfies all the conditions? Please note that this is fine if this is the case, I just want to know whether this is the case or not.

---

> > > ### Author Response · Authors · 2022-08-04
> > > **Response to reviewer x58D**
> > >
> > > Thank you for your sharp point on the part. In short, we presented numerical examples as one of the numerical evidence to support our theory.
> > >
> > > Our research deals with sufficient conditions to achieve (nearly) optimal posterior consistency. In other words, it was shown that the model satisfying the conditions of the paper had posterior consistency. As mentioned in the revised submission, our conditions are harsh as we have to consider a fairly complex model. For instance, a sufficient conditions for estimating the functions in the experiments are that, according to Theorem, the depth $L_n$ of the neural network model should be greater than 30, the width $W_n$ greater than 400, and proper Gaussian mixture prior. We tried to show through numerical experiments that a suitable neural network has sufficiently good theoretical properties even if the theoretical conditions are partially satisfied. In experiments, we showed that valid inference is possible by using the prior as it is, and considering the simple models which have a depth of 5 and a width of 200. In numerical experiments, we wanted to explain the tendency of theoretical results and to show the meaning of the word 'practical' in our research.

---

> > > > ### Comment · Reviewer_x58D · 2022-08-09
> > > > **Thank you for clarification**
> > > >
> > > > Thank you very much for this clarification. It is clear now.

---

### Official Review · Reviewer_T5q1 · 2022-07-07

**Rating:** 7
**Confidence:** 4
**Soundness:** 3 good
**Presentation:** 4 excellent
**Contribution:** 3 good

**Summary:**

This paper present a proof of Bayesian posterior contraction rate in Besov spaces for Bayesian neural networks priors for the regression problem on $[0,1]^d$. More specifically, two priors are presented : spike and slab and shrinkage (the latter being easier to implement). Under the frequentist assumption that the true regression function $f_0$ belongs in a Besov space $B_{p,q }^s$, it is shown that the two priors achieve minimax rates of convergence (up to log terms) with a (non adaptive) choice of prior parameters (which includes a choice of depth, width & sparsity of the network). Moreover, it is shown that an adaptive rate of convergence over all smoothness index $s>0$ is possible for a spike and slab prior with an additional prior on the architecture. This work extend previous results for Bayesian posterior contraction in Hölder spaces and frequentist minimax theory in Besov for neural networks.

**Questions:**

I have two questions :

- in appendix C1, proof of lemma 3 (35) : can you quickly justify how you lower bound the norm ?

- in the proof of theorem 1 (and subsequently in the paper for the other analog proofs), it is not clear how the prior on the connectivity pattern (ie on $gamma$) appears in the proof of the prior thickness result (43)-(44)-(45), should there be a $\binom{T_n}{S_n}$ factor somewhere or am I making a mistake ?

**Limitations:**

From my point of view the main limitation of the paper is that adaptive result is only proven in the case of the spika and slab prior, and that the numerical expriments are conducted on toy examples with known smoothness levels. It could be interesting to, at least, discuss why you are only presenting an adaptive result for the first prior.

**Strengths And Weaknesses:**

This paper is a strong theoretical results closing a gap for posterior contraction of BNN in Besov spaces. Concerning the priors, it presents a new (computationally more efficient) shrinkage prior that is shown to lead to good posterior contraction rates, which is (to the best of my knownledge) a new result even for BNN in Hölder spaces.
Moreover, I think that paper is overall wery clear and well written. However,

---

> ### Author Response · Authors · 2022-08-02
> **Reply to reviewer T5q1**
>
> Thanks for your comments and suggestions. Overall, there were few problems in the mathematical proof, but the results were unchanged. We found some issues caused by the ambiguous notation, and fixed all of them.
>
> ## Q.
>
> > in appendix C1, proof of lemma 3 (35) : can you quickly justify how you lower bound the norm?
>
> Is it correct to ask the process of calculating the **upper bound** in the inequality of Lemma 3 (35)?
>
> Since there were notations to be fixed in the existing submission, it has been corrected as follows.
>
> $$\lVert A_k^+(f)(x) \rVert_{\infty} \leq \max_j  \lVert W_{j,:}^{(k-1)} \rVert_1 \lVert A_{k-1}^+(f)(x) \rVert_{\infty}  + \lVert b^{(k-1)} \rVert_{\infty}$$
>
> $$\leq WB\lVert A_{k-1}^+(f)(x) \rVert_{\infty} + B$$
>
> $$\leq (W+1)(B \vee 1) \lVert A_{k-1}^+(f)(x) \rVert_{\infty} $$
>
> $$\leq (W+1)^{k-1}(B \vee 1)^{k-1},$$
>
> for all $x$, where $A_{j, :}$ is the $j$-th row of the matrix $A$.
>
> If your question is about this part, we got the above inequality as follows:
>
> First, from the triangular inequality,
>
> $$ \lVert Ax + b \rVert_\infty \leq \lVert Ax \rVert_\infty + \lVert b \rVert_\infty.$$
>
> Next, by the definition of the matrix norm and the properties of the infinity norm of the matrix, we got
> $$\lVert Ax\rVert_\infty \leq \lVert A \rVert_\infty \lVert x \rVert_\infty =  \max_j \lVert A_{j,:}\rVert_1 \lVert x \rVert_\infty.$$
>
> After the second inequality, we used the conditions of the parametric space and mathematical induction for the last inequality.
>
> ## Q.
>
> > in the proof of theorem 1 (and subsequently in the paper for the other analog proofs), it is not clear how the prior on the connectivity pattern (ie on $\gamma$) appears in the proof of the prior thickness result (43)-(44)-(45), should there be a $\binom{T_n}{S_n}$ factor somewhere or am I making a mistake ?
>
> You are right. It was our mistake. We correct the proof in the revised paper. We confirmed that the results of theorem did not change. Sorry for the confusion and thank you for correcting the proof.
>
> ## Q.
>
> > It could be interesting to, at least, discuss why you are only presenting an adaptive result for the first prior.
>
> In Theorem 3, we mentioned the conditions that the shrinkage prior must satisfy. As you can see, conditions such as tail probability depend on the parameters $L_n, ~W_n$, and $B_n$. For adaptive estimation like Theorem 2, it is necessary to propose a general shrinkage prior distribution that satisfies all conditions even when $L_n, ~W_n$, and $B_n$ are varying, but this has not been solved yet. We have planned this in the future work.

---

### Official Review · Reviewer_Z4is · 2022-07-11

**Rating:** 5
**Confidence:** 3
**Soundness:** 3 good
**Presentation:** 2 fair
**Contribution:** 2 fair

**Summary:**

This work extends the results of Polson and Rocková [2018] on the minimax contraction rate of Bayesian ReLU network from the Hölder space to the more general Besov space. Such result brings more freedom to the underlying functional form behind the data with similar minimax contraction rate guarantee. Beside the spike-and-slab prior, the result with shrinkage prior is also given but with more computational efficiency.

**Questions:**

The underlying functions are usually assumed to be continues when using BNN for real-data modelling. Since the advantage of Besov space is its ability to include non-differentiable functions, can you find one or some real data that needs the functions from Besov space rather than Hölder space or even L^p space?

Can you highlight the challenges on the proposed extensions?

**Strengths And Weaknesses:**

The strong point of this work is the ability to extend the functional form modelled by Bayesian ReLU network and its posterior contraction rate guarantee. The motivation is clear, and the idea is well presented.

However, as the authors said, this is an extension of existing works Polson and Rocková [2018] with similar backgrounds, settings, and results, and the derivative looks a little straightforward, so the innovation is limited. The authors did not highlight the challenges or difficulties in how to extend the results from the Hölder space to the more general Besov space.

Another main weakness is the evaluation. Only one simple example is given to show the ability of designed prior for BNN regression. It would be better to compare with other options, like iid Gaussians, spike-and-slab, and the Gaussians priors with learnable hyperparameters. In practice, we normally learn the parameters of priors from the data as well rather than just fixing them while we using BNN for data modelling. Is there any effect from such learnable priors?

The underlying functions are usually assumed to be continues when using BNN for real-data modelling. Since the advantage of Besov space is its ability to include non-differentiable functions, can you find one or some real data that needs the functions from Besov space rather than Hölder space or even L^p space?

Some minor problems include: the symbols should be explained, like “A” in (9), any conflict between “L_1” in Line 108 and “L_n” in (10), Line 158, “a expectation” and Line 185, “acontinuous”

---

> ### Author Response · Authors · 2022-08-02
> **Reply to reviewer Z4is**
>
>
> Thanks for your comments and suggestions.
>
> ## Q.
>
> > The authors did not highlight the challenges or difficulties in how to extend the results from the Hölder space to the more general Besov space.
>
> > Can you highlight the challenges on the proposed extensions?
>
> Difficult points when considering the Besov space compared to the Hölder space were the following.
>
> 1. The shape of the space was not intuitive. We could see easily that it was a larger (general) space, including the Hölder space "by the formula", but it was not that easy to think of which functions were "actually" included. For this reason, we have mentioned in the paper example functions (including $f_1$ and $f_2$) that are easy to understand for the Besov space.
> 2. To show posterior consistency using the Lemma in Ghosal and Van Der Vaart (2007), it was necessary to prove the upper bound of entropy and the lower bound of the prior mass. Of these, the latter is affected by the enlarged function space. From Suzuki (2018), we found an empirical minimizer close enough to the true function which in the model space, and computed the lower bound of prior mass using norm inequalities. In fact, it was difficult to found the necessary conditions of priors for the theorems. We checked the related works and found the conditions mentioned in the paper: (1) enough mass around zero, (2) thick tail to sample true function (3) while not too much.
>
> ## Q.
>
> > It would be better to compare with other options, like iid Gaussians, spike-and-slab, and the Gaussians priors with learnable hyperparameters.
>
> > In practice, we normally learn the parameters of priors from the data as well rather than just fixing them while we using BNN for data modelling. Is there any effect from such learnable priors?
>
> We added the result of using the Gaussian prior and the Gaussian mixture prior proposed in the revised submission for simple synthetic data. Both model give similar results, but the Gaussian prior at $f_1$ and the Gaussian mixture prior at $f_2$ give slightly better results.
>
> Note: As replied to Reviewer x58D, it is an experimental result for a smaller model which does not satisfiy the conditions of the paper.
>
> ## Q.
>
> > Since the advantage of Besov space is its ability to include non-differentiable functions, can you find one or some real data that needs the functions from Besov space rather than Hölder space or even L^p space?
>
> When we first planned the study, we were interested in 'why does an artificial neural network model (a machine learning model) outperform a (traditional) statistical model?'. One of the answers was 'what if the relationship between explanatory and response data is in the form of an indifferentiable function?'. Of course, it is difficult to verify this in practice. This is because it is almost difficult to confirm the form of specific functions of explanatory variables and response variables in real data. However, empirically, we know that there are fields in which decision-tree-based model such random forest or neural network models perform better than models that find smooth functional relationships such as Kernel regression. For these data, it is natural that there would be a non-smooth relationship between the explanatory variable and the response variable. For this reason, we think that it is essential to compare the relationship between other models and deep learning models for follow-up studies.
>
> In short, we speculate that there is a non-smooth relationship in the recent machine learning problems (image processing, natural language processing, etc.) where neural network models perform better than traditional statistical models.
>
> ## Q.
>
> > Some minor problems include: the symbols should be explained, like “A” in (9), any conflict between “L_1” in Line 108 and “L_n” in (10), Line 158, “a expectation” and Line 185, “acontinuous”
>
> Thanks for your detailed comments. We have corrected everything you mentioned in the revised submission.

---

> > ### Comment · Reviewer_Z4is · 2022-08-08
> > **response to authors**
> >
> > Thanks for your responses!
> >
> > The question about the learnable hyperparameters is about the prior derived in the paper, which seems to be a fixed prior. However, we normally set a learnable prior to practice. Will this learnable prior be better than your derived one?
> >
> > Neural network-based method may also implicitly assume an underlying smooth function, so we can draw the conclusion that the better performance of neural network-based methods indicates the underlying non-smooth function. The authors may compare your methods with the old one (within the Besov space and Hölder space separately) on the real-world datasets in a fair setting to see the different performances. If the method within Besov space can achieve better performance in some datasets, we may know the non-smooth function approximation may be beneficial.
> >
> > All in all, I suggest the authors to do more experiments on real datasets to show potential applications.

---

> > > ### Author Response · Authors · 2022-08-09
> > > **Response to reviewer Z4is**
> > >
> > > Thanks for your constructive questions.
> > >
> > > ## Q.
> > >
> > > > The question about the learnable hyperparameters is about the prior derived in the paper, which seems to be a fixed prior. However, we normally set a learnable prior to practice. Will this learnable prior be better than your derived one?
> > >
> > > We think it would be nice to consider a learnable prior that estimates the hyperparameters of the prior as you said. For instance, we can assume the half-Cauchy distribution of the standard deviation of the parameters. However, there is a problem in that the amount of computation increases, and in the current submission, good enough results were obtained with a simpler fixed prior. We will consider the theoretical study and actual comparison of the learnable prior. Thanks for the suggestion.
> > >
> > > ## Q.
> > >
> > > > All in all, I suggest the authors to do more experiments on real datasets to show potential applications.
> > >
> > > Thank you for your good suggestions. We will consider experiments that can apply the models we proposed to real data.

---

### Author Response · Authors · 2022-08-02
**Revision has been uploaded**

Dear reviewers,

Thanks for the constructive and detailed feedback. Based on your review, we have revised the paper as follows:

* The value of the constant $c_{(d, m)}$ which determines the complexity of the neural network model, was corrected by checking the references. Accordingly, the numerical experiments in the submitted paper are also modified. The results related to the numerical experiments were moved to the appendix based on the advice of reviewer x58D.
* We changed the algorithm in numerical examples to the NUTS algorithm from the variational approximation.
* As pointed out by reviewer T5q1, we have corrected the proof of the Theorem 1. We confirmed that there were no problems with the result of the theorem.
* By all of your helpful feedback, we have corrected ambiguous and incorrect expressions in spelling, notation, and the proofs. Especially, we rewrite the abstract and conclusion more clearly.

---

### Meta-Review · Area_Chair_8Lxw · 2022-08-25

**Recommendation:** Accept
**Confidence:** Less certain

**Metareview:**

This work conducts novelty study and extends the results on asymptotic convergence of Bayesian ReLU networks from the Hölder space to the more general Besov space. The reviewers consider it "a strong theoretical results closing a gap for posterior contraction of BNN in Besov spaces".

The authors' feedback addressed a few concerns in the initial reviews including the lack of clarity, the question on the technical challenges to extend to a Besov space, and an error in the proof.

During the author-reviewer discussion period, the authors also corrected a constant which determines the complexity of the neural network model. As a result, the condition of the theory became "harsh" (see authors' response to reviewer x58D), and the numerical results did not satisify the theory's requirement any more. The authors provided an updated version of the paper to include the change and moved the experiments to appendix. Nonetheless, the reviewers did not think that change decreased its theoretical value and still considered it above the threshold for acceptance due to its novelty.

The remaining concerns from the reviewer is lack of evaluations for the non-smoothness assumption and its usage in some real applications, also for the possible difference between a purposely designed fixed prior with a learnable prior.

**Award:**

No

---

### Decision · Program_Chairs · 2022-09-14

Accept